# Myo-Inositol Reverses TGF-β1-Induced EMT in MCF-10A Non-Tumorigenic Breast Cells

**DOI:** 10.3390/cancers15082317

**Published:** 2023-04-15

**Authors:** Noemi Monti, Simona Dinicola, Alessandro Querqui, Gianmarco Fabrizi, Valeria Fedeli, Luisa Gesualdi, Angela Catizone, Vittorio Unfer, Mariano Bizzarri

**Affiliations:** 1Department of Experimental Medicine, Sapienza University of Rome, Viale Regina Elena 324, 00161 Rome, Italy; 2Systems Biology Group Laboratory, Sapienza University, 00161 Rome, Italy; 3Section of Histology and Embryology, Department of Anatomy, Histology, Forensic Medicine and Orthopedics, Sapienza University of Rome, Viale Regina Elena 336, 00161 Rome, Italy; 4The Experts Group on Inositol in Basic and Clinical Research (EGOI), 00161 Rome, Italy; 5Gynecology Department, UniCamillus—Saint Camillus International University of Health and Medical Sciences, 00161 Rome, Italy

**Keywords:** EMT, TGF-β1, myo-Ins, cytoskeleton, MET, E-cadherin

## Abstract

**Simple Summary:**

Inflammatory conditions can enact the emergence of cancer, especially by promoting an epithelial-mesenchymal transition (EMT). Furthermore, EMT is a critical requirement for the dissemination of cancerous cells. The discovery of pharmacological agents able to inhibit EMT has become a critical issue in recent times. Herein we demonstrate that myo-inositol (myo-Ins), can efficiently rescue normal breast cells committed to an inflammatory phenotype upon the addition of the inflammatory TGF-β1 stimulus. Myo-Ins was able to almost completely inhibit the resulting invasive–migrating phenotype, namely by reverting EMT. A critical step is the reconstitution of proper E-cadherin-based cell-to-cell junctions, which are instrumental in the recovery of a tissue-like structure. Moreover, myo-Ins re-established a normal gene expression pattern, while normalizing the microenvironment by reducing collagen and metalloproteinase release. These results highlight the relevance of inflammation in promoting a precancerous state and provide useful new perspectives in cancer treatment and prevention with natural compounds.

**Abstract:**

Epithelial-Mesenchymal Transition (EMT), triggered by external and internal cues in several physiological and pathological conditions, elicits the transformation of epithelial cells into a mesenchymal-like phenotype. During EMT, epithelial cells lose cell-to-cell contact and acquire unusual motility/invasive capabilities. The associated architectural and functional changes destabilize the epithelial layer consistency, allowing cells to migrate and invade the surrounding tissues. EMT is a critical step in the progression of inflammation and cancer, often sustained by a main driving factor as the transforming growth factor-β1 (TGF-β1). Antagonizing EMT has recently gained momentum as an attractive issue in cancer treatment and metastasis prevention. Herein, we demonstrate the capability of myo-inositol (myo-Ins) to revert the EMT process induced by TGF-β1 on MCF-10A breast cells. Upon TGF-β1 addition, cells underwent a dramatic phenotypic transformation, as witnessed by structural (disappearance of the E-cadherin–β-catenin complexes and the emergence of a mesenchymal shape) and molecular modifications (increase in N-cadherin, Snai1, and vimentin), including the release of increased collagen and fibronectin. However, following myo-Ins, those changes were almost completely reverted. Inositol promotes the reconstitution of E-cadherin–β-catenin complexes, decreasing the expression of genes involved in EMT, while promoting the re-expression of epithelial genes (keratin-18 and E-cadherin). Noticeably, myo-Ins efficiently inhibits the invasiveness and migrating capability of TGF-β1 treated cells, also reducing the release of metalloproteinase (MMP-9) altogether with collagen synthesis, allowing for the re-establishment of appropriate cell-to-cell junctions, ultimately leading the cell layer back towards a more compact state. Inositol effects were nullified by previous treatment with an siRNA construct to inhibit CDH1 transcripts and, hence, E-cadherin synthesis. This finding suggests that the reconstitution of E-cadherin complexes is an irreplaceable step in the inositol-induced reversion of EMT. Overall, such a result advocates for the useful role of myo-Ins in cancer treatment.

## 1. Introduction

Epithelial-mesenchymal transition (EMT) is a transdifferentiating, reversible process involved in cell phenotypic changes [1]. This transition occurs in physiological conditions—embryogenesis and tissue repair [2], albeit EMT can be activated by cells during tumor development or fibrotic processes [3].

In normal conditions, epithelial cells interact with each other through cell–cell junctions, creating a monolayer sheet. However, cells headed toward EMT modify their cytoskeletal setup, as well as the distribution of junctional complexes, especially those based on E-cadherin and β-catenin interaction [4]. Noticeably, E-cadherin is responsible for maintaining lateral contacts between cells via the adherents’ junctions and cell adhesion. When junctional structures are either weakened or lost, cells disaggregate and become more scattered, undergoing morphological alterations. Consequently, cells proceed in acquiring a mesenchymal phenotype by gaining mesenchymal markers and capabilities [5]. Moreover, cells display biochemical changes, including enhanced migratory capacity, invasiveness, and increased production of extracellular matrix (ECM) components [6]. It is noteworthy that EMT can be triggered by both extracellular and intracellular stimuli, involving several molecular pathways, finally leading to a reorganization of the overall phenotypic asset. Transforming growth factor-β1 (TGF-β1), a ubiquitously expressed cytokine, is a major biochemical inducer of EMT, playing an important role in a variety of biological processes [7]. TGF-β1 can act either through a very well-described canonical signaling pathway (involving Smad2 and Smad3) or by a wide variety of Smad-independent pathways (known as noncanonical signaling) to modify cell function [8].

Remarkably, EMT is associated with chronic inflammation [9]. Chronic inflammation strongly disrupts the cell-microenvironment cross talk, by involving several actors, including epithelial cells, ECM, and many other stroma components. Generally, inflammation evolves towards healing to achieve a near complete restitutio ad integrum. However, in some cases, inflammation might lead to either a fibrotic reaction or a cancer transformation [10,11]. It is worthy of note that EMT associated with the inflammatory process is considered a key step for tumor invasion and metastasis [12]. In both cases, the inflammation acts primarily by remodeling the microenvironment architecture and composition.

Nevertheless, EMT is not irreversible and can be “reversed” through an inverse process, the mesenchymal–epithelial transition (MET)—which plays an important role in tissue repair and morphogenesis, as well as in several pathological conditions, such as tumor progression or fibrosis [13,14]. Notably, MET has been observed as occurring either spontaneously or after treatment in those cancers that undergo differentiation or were committed to regression [13,15]. A very special case is represented by MET observed during tumor reversion, as described by a number of reports [16,17].

In order to test the capability of natural pharmacological agents in counteracting TGF-β1 induced EMT, we investigated the effects of inositol upon the immortalized line of normal breast cells—MCF10A—previously stimulated with TGF-β1 to elicit an EMT response. Myo-Inositol (myo-Ins) is a natural polyol [18], which has already been shown to exert an appreciable anti-inflammatory activity in human beings [19], as well as significant antitumor activities, leading to MET [20]. Our results demonstrated that TGF-β1 was successful in promoting the epithelial to mesenchymal transition in MCF-10A cells that acquired impressive migrating and invasive properties. As expected, myo-Ins inhibited almost all the functional and biochemical changes triggered by TGF-β1, while promoting a phenotypic reversion, leading cells to recover their native phenotype through a mesenchymal–epithelial transition.

## 2. Materials and Methods

### 2.1. Experimental Cell Model and Treatments

The nontumorigenic breast epithelial cell line MCF-10A (ATCCCRL-10317), was seeded into 100 mm dishes in a DMEM-F12 Medium (1:1) supplemented with 5% horse serum, 20 ng/mL EGF (epidermal growth factor), 10 mg/mL insulin, cholera toxin (0.5 mg/mL), hydrocortisone (50 μM), and antibiotics (penicillin 100 IU/mL and streptomycin 100 μg/mL). Cells were cultured at 37 °C in an atmosphere of 5% CO_2_ in the air. The medium was changed every third day. At confluence, cells were sub-cultured after removal with 0.05% trypsin and 0.01% EDTA, for a maximum of thirty passages. MCF-10A cells were firstly treated with 10 ng/mL TGF-β1 (PeproTech #100-21, Cranbury, NJ, USA) for five days and then, on the fifth day, 4 mM of myo-Ins (ANDROSITOLs LAB; Lo. Li. Pharma s.r.l., Rome, Italy) were added in a complete medium either for one hour or for six hours to analyze gene expression and protein expression, respectively.

### 2.2. RNA Extraction and RT-qPCR

Total RNA isolation from cells, untreated and treated with TGF-β1, with and without myo-Ins, was performed using TriReagentR (Cat #T9424 Sigma-Aldrich, Saint Luis, MC, USA) following the manufacturer’s instructions. The nucleic acid concentration was determined through Nanodrop measurements. The same amount of total RNA was reverse transcribed using an Optifast cDNA Synthesis Kit. Quantitative PCR (qPCR) has been performed with an iTaq Universal SYBRR Green. Levels of mRNA were standardized using GAPDH (Hs_GAPDH_1_SG; Quanti Tech Primer Assay). The analysis was performed in triplicate for each sample. Primers used in qPCR are listed in Appendix A (Appendix A).

### 2.3. Western Blot Assay

After treatment with TGF-β1 and myo-Ins for six hours, MCF-10A cells were exposed to cold PBS and then frozen at −80 °C for almost 24 h. Proteins proteins were extracted using a RIPA lysis buffer (Sigma-Aldrich) with a mix of protease and phosphatase inhibitors (Complete-Mini Protease Inhibitor Cocktail Tablets, Roche, Mannheim, Germany; PhosSTOP inhibitor tablets for phosphatase, Roche, Mannheim, Germany). Lysates were centrifuged at 8000× *g* at 4° C for 10 min to collect supernatants. The protein content was measured according to Bradford’s method. Protein lysates were separated on SDS-polyacrylamide gels and blotted onto nitrocellulose membranes (BIO-RAD, Bio Rad Laboratories, Hercules, CA, USA). After blocking with nonfat milk 5% in PBS-T 0.1%, membranes were incubated with primary antibodies at 4 °C overnight, washed 3 times with PBS-T 0.1%, and then incubated with antimouse or antirabbit HRP-conjugated antibodies for 1 h at room temperature. Primary antibodies were β-catenin (sc-7963) N-cadherin (sc-7939), vimentin (sc-6260) (Santa Cruz Biotechnology, Dallas, TX, USA), keratin18 (#4548), E-cadherin (#3195), and GAPDH (#2118) (Cell Signaling Technology, Danvers, MA, USA). Antigens were detected through chemiluminescence using an ECL kit (Western Bright ECL HRP Substrate, Waltham, MA, USA; Immunological science), according to the manufacturer’s instructions. Optical density was acquired using Chemidoc c300 (Azure Biosystems, Dublin, CA, USA). All the Western blot experiments were performed three times.

### 2.4. Confocal Microscopy

MCF-10A cells were seeded in a complete medium in 8-well μ-slides (Ibidi GmbH, Am Klopferspitz 19, D-82152 Martinsried, Germany) at the concentration of 1.5 × 103 cells for each well. The four conditions for the experiment were as follows: untreated control; myo-Ins; TGF-β1; TGF-β1 + myo-Ins. After incubation with TGF-β1 and/or myo-Ins, cells were fixed with 4% paraformaldehyde for 10 min at 4 °C, washed twice with PBS, and then permeabilized for 60 min using PBS, 2% BSA, 0.1% Triton X-100 (Sigma-Aldrich) followed by anti-β-catenin (Santa Cruz Technology) and anti-E-cadherin (Cell Signaling Technology) staining in PBS, 2% BSA at 4 °C overnight. Cells were washed with PBS and incubated for 1 h at room temperature with the appropriate secondary antibody (FITC and TRITC conjugated, Invitrogen Molecular Probes, Eugene, OR, USA). Negative controls were managed in the same conditions, aside from primary antibody staining. Nuclei were stained with TO-PRO-3 (Thermo Fisher Scientific, Waltham, MA, USA) in PBS for 15 min in the dark. For F-actin visualization, after fixation in 4% of paraformaldehyde for ten minutes at 4 °C, cells were permeabilized for ten minutes at 4 °C, with an equimolar cold solution of 1:1 ethanol–acetone. Then, the cells were washed with PBS and incubated with Rhodamine Phalloidin (Invitrogen Molecular Probes Eugene, Eugene, OR, USA) for 25 min in the dark. Cells were then washed in PBS, and mounted in buffered glycerol (0.1 M, pH 9.5). Finally, fluorescent labeling was analyzed using a Leica Confocal Microscope TCS SP2 (Leica Microsystems Heidelberg GmbH, Mannheim, Germany), using a 488, 543, and 633 nm exciting laser for FITC, TRITC e TO-PRO-3 fluorophores. The images were obtained at 20× and 40× magnifications.

### 2.5. Migration and Invasion Assays

Samples of 2.5 × 104 cells, control (CTRL), and stimulated with either 10 ng/mL TGF-β1, 4 mM myo-Ins as a single agent, or with both TGF-β1 and myo-Ins, were placed in 500 µL of DMEM-F12 supplemented with 0.1% horse serum in the upper side of 8-µm filters (Falcon, BD Biosciences, San Jose, CA, USA) (upper chamber), and placed in the wells of a 24-well (lower chamber), containing 0.8 mL of DMEM F12 + 5% horse serum. After 24 h of incubation, the migratory cells on the lower surface of membranes were fixed and stained with Hemacolor^®^ (HX54775574, Merck, Darmstadt, Germany) following the manufacturer’s instructions. Cell migration was determined by counting the number of cells on the membranes in at least 4–5 randomly selected fields using a Zeiss LSM 900 Confocal Microscope Airyscan 1 (Oberkochen, Germany) optical microscope. For the invasion assay, 2.5 × 104 cells were seeded in 500 µL of DMEM-F12 supplemented with 0.1% of horse serum on the upper surface of the transwells containing Matrigel (BD Bio-CoatTM growth factor reduced MATRIGEL TM invasion chamber, BD Bioscences-Discovery Labware, two Oak Park, Bedford, MA, USA). Transwells were located on a 24-well plate (Falcon, BD Biosciences) containing 0.8 mL of the same culture medium, supplemented with 5% horse serum used as a chemoattractant. After 24 h of incubation, Matrigel invading cells were stopped, fixed, and stained with Hemacolor^®^ (following the manufacturer’s instructions), and then analyzed by optical microscope (Zeiss LSM 900 Confocal Microscope Airyscan 1). The cellular invasion was determined by counting the number of cells on the membranes in at least 4–5 randomly selected fields using a Zeiss Zeiss LSM 900 Confocal Microscope Airyscan 1 optical microscope.

### 2.6. Zymography

The enzymatic activity of MMP-9 was determined by gelatin zymography. Conditioned media of the control cells and TGF-β1 and 4 mM myo-Ins alone and/or together treated cells were prepared with a standard SDS–poly-Acrylamide gel loading buffer containing 0.01% SDS without β-mercaptoethanol and not boiled before loading. Then, samples were subjected to electrophoresis with 12% SDS–PAGE containing 1% gelatin. Gels were then washed twice with distilled water containing 2.5% Triton-X100 for 30 min at room temperature and incubated in collagenase buffer (0.5 m Tris-HCl pH7.5, 50 mM CaCl2 and 2 m NaCl) overnight at 37 °C, stained with Coomassie brilliant blue R-250 and treated with destaining solution (30% methanol, 10% acetic acid, and 60% water). The MMP gelatin zymography was performed three times.

### 2.7. E-Cadherin Interference

Transient transfections were performed using Lipofectamine™ RNAiMAX, Invitrogen™, and siRNA of CDH1 (E-cadherin) (Ambion™ by Life Technologies, Carlsbad, CA, USA) in 250 μL of Opti-MEM™ (Gibco™) according to manufacturer’s instructions. Then, 250 μL of lipofectamine and siRNA were added at T0 directly in the cell-culture medium, and after 1 h, TGF-β1 was added. Upon five days of TGF-β1 and siRNA treatment, 4 mM of myo-Ins was added for six hours. The process was stopped at 120 h of transfection. Further, β-catenin and E-cadherin were investigated with confocal microscopy as previously described.

### 2.8. Statistical Analysis

Statistical analysis was performed by one-way ANOVA and Turkey’s multiple comparison test using GraphPad Prism 8. The standard error of the mean and *p* values of <0.05 were considered statistically significant.

## 3. Results

### 3.1. TGF-β1 Induces Morphological Changes in MCF-10A Cells

Cells treated for five continuous days with TGF-β1 undergo EMT. Under physiological conditions, MCF-10A epithelial cells have an elongated cuboidal shape, with an apicobasal polarity, and the cells strictly interact with each other through the adherent junctions (Figure 1A,a). However, the addition of TGF-β1 leads to morphological changes in MCF-10A. Cells assume a spindle-like shape, a distinctive feature of fibroblasts or mesenchymal cells, and progressively lose the adherent junctions. Conclusively, cells become scattered and no longer form a physiological epithelial sheet (Figure 1B,b).

### 3.2. Modulation of Gene Expression during EMT

The transition process induced by TGF-β1 not only affects the morphological feature. The phenotypic modification is accompanied by an altered expression of EMT parameters. EMT is characterized by the progressive loss of epithelial markers while acquiring mesenchymal features. Therefore, we analyzed the modulation of EMT parameters upon continuous induction with TGF-β1 at different time points, from every 24 h up to five days. We observed an opposite trend in the expression of the two main markers involved in the transition: E-cadherin decreases while N-cadherin increases (Figure 2A,B). The early increase in E-cadherin expression during the first two days following TGF-β1 induction, albeit significant, can be viewed as a transitory, adaptive reaction manifested by cells trying to counteract the occurring transition. Moreover, we analyzed the expression of Snai1 and PI3K, two factors involved in the triggering of EMT [21,22]. Snai1 represses the epithelial phenotype mostly by inhibiting the expression of E-cadherin [23]. In turn, PI3K directly regulates the expression of Snai1. We observed that the expression of these factors noticeably increases upon continuous induction with TGF-β1 (Figure 2C,D). We highlighted a similar trend by analyzing collagen I and fibronectin, two main components of ECM [24]. The expression of both parameters extremely increases after 72 h upon TGF-β1 addition (Figure 2E,F). This finding clearly suggests that the epithelial cells have been committed toward an EMT. Unexpectedly, continuous treatment with TGF-β1 activates an autocrine positive loop that ultimately leads to an increased synthesis of endogenous TGF-β1 (Figure 2G). Such a result suggests that EMT induction through TGF-β1 can lead to a self-sustained process, reinforcing the mesenchymal transition.

### 3.3. Myo-Ins Counteracts TGF-β1-Dependent Effects on EMT Marker Expressions

The aforementioned altered gene expression is almost completely nullified following the addition of myo-Ins after five days of treatment with TGF-β1. Noticeably, this reversion is an early effect of inositol, as changes occur even only one hour after myo-Ins treatment. Furthermore, cells treated with myo-Ins show an increased expression of E-cadherin, while N-cadherin levels decrease (Figure 3A,B). Furthermore, the expression of Snai1 and PI3K was significantly decreased by the addition of myo-Ins (Figure 3C,D). It is noteworthy that myo-Ins increases PI3K expression in normal cells while reducing the same marker when cells are stimulated by TGF-β1. This result suggests that myo-Ins specifically modulates PI3K levels when this enzymatic pathway is challenged by an inflammatory-like stimulus. Consequently, myo-Ins—rather than being a simple “inhibitor”—could be considered a “homeostatic regulator” of the PI3K function.

Myo-Ins induces a marked decrease in collagen I expression while the expression of fibronectin was unaffected (Figure 3E,F). This finding strongly confirms that myo-Ins can efficiently counteract a critical step during the EMT process. Conversely, it is worthy of note that myo-Ins treatment was ineffective in downregulating the increase in TGF-β expression we observed in cells stimulated by exogenous TGF-β1 (Figure 3G). This finding seems to indicate that myo-Ins can antagonize only some TGF-β1 induced effects upon the cell machinery.

### 3.4. F-Actin Remodeling

Epithelial cells undergoing EMT assume a typically mesenchymal-like shape and lose their apico–basal polarity [25]. TGF-β1 fosters a significant remodeling of cortical F-actin, given that cells undergoing EMT show an almost uniform redistribution of actin while losing their polarity. Moreover, the presence of stress fibers increases significantly (Figure 4C,c). In addition, we observed that those cells displayed protruding edges from the cell surface. Cells adopt a characteristic mesenchymal, sharp shape, displaying several kinds of invading structures (invadopodia, lamellipodia, and pseudopodia). Whereas myo-Ins does not influence actin distribution in the control samples (Figure 4A,a), inositol almost completely reverts F-actin changes induced by TGF-β1 (Figure 4B,b). Additionally, TGF-β1 fosters the organization of F-actin in stress fibers, while inositol inhibited the emergence of both stress fibers and lamellipodia/invadopodia (Figure 4D,d).

### 3.5. Myo-Ins Almost Completely Inhibits the Migrating and Invasive Capabilities Induced by TGF-β1

During EMT, epithelial cells acquire invasiveness and migration ability. Migration and invasion assays were performed on MCF-10A cells, stimulated by TGF-β1 with and without myo-Ins. The TGF-β1 induction enhances the number of cells able to migrate, compared to the control cells and compared to those treated with myo-Ins alone (Figure 5A). This feature is associated with an increased invasive profile. The invasion assay demonstrated that MCF-10A cells stimulated with TGF-β1 acquired an invasive capacity compared to the control cells and to those treated with myo-Ins alone (Figure 5B) (Appendix A). The migratory and invasiveness ability of EMT-induced epithelial cells is partly supported by the proteolytic activity of metalloproteinases, that degrade the components of the extracellular matrix. It is noteworthy that MMP-9 shows increased activity in MCF-10A cells treated with TGF-β1 compared to the control cells and to those treated with myo-Ins alone. As expected, MMP-9 activity decreases when myo-Ins was added following TGF-β1 induction (Figure 6).

### 3.6. Myo-Ins Antagonizes TGF-β1-Dependent Effects on CSK Rearrangement

The cytoskeleton (CSK) plays a vital role in sustaining the integrity of epithelial cells [26]. Epithelial cells mainly exhibit cytokeratin in their CSK, whereas mesenchymal cells display increased levels of vimentin. During the EMT, in epithelial cells, cytokeratin is replaced by vimentin. We analyzed CSK parameters involved in EMT by Western blot in MCF-10A treated with TGF-β1 for five days and then with myo-Ins for six hours. As expected, keratin-18 and vimentin were highly modulated in an opposite way in normal cells, while TGF-β1 significantly decreases keratin-18 expression and increases vimentin release (Figure 7A,B). However, myo-Ins reverts the picture by decreasing vimentin and raising keratin-18 expression. Moreover, myo-Ins dramatically counteracts the changes induced by TGF-β1-induced EMT upon cadherins. Whereas N-cadherin was significantly downregulated, E-cadherin levels increased up to two-fold with respect to the values observed in TGF-β1 treated cells (Figure 8A,B). Similarly, myo-Ins induced a marked rise in β-catenin levels (Figure 8C).

### 3.7. Myo-Ins Restores Physiological E-Cadherin/β-Catenin Distribution

Myo-Ins not only influences the expression pattern of E-cadherin and β-catenin. It presumably also influences their cytosolic distribution. Hence, we performed confocal microscopy analysis to investigate the localization of E-cadherin and β-catenin in MCF-10A (Figure 9). Cell–cell junctions are critical in maintaining cell polarity and integrity. E-cadherin/β-catenin complex plays an important role in maintaining epithelial integrity, and the disassembly of the cadherin–catenin association plays a pivotal role in changing cell morphology and functions. Upon TGF-β1 treatment, breast cells lose cell-to-cell junctions, and are scattered and detached from each other. E-cadherin–β-catenin complexes are significantly reduced, while β-catenin translocates in the cytosol and into the nucleus. On the contrary, myo-Ins promotes the reconstitution of cell-to-cell junctions and favors the de novo re-establishment of E-cadherin–β-catenin complexes. Under myo-Ins treatment, cells were no longer detached and dispersed across the dish plate, recovering a tissue-like appearance with well-established cell-to-cell adhesions. Overall, myo-Ins was effective in restoring the connectivity among cells, and in promoting the reconstitution of a tissue-like structure.

### 3.8. E-Cadherin Silencing

According to the aforementioned results, we hypothesized that the changes in E-cadherin/β-catenin induced by myo-Ins could constitute the critical step in reversing EMT induced by TGF-β1 stimulation. Therefore, we replicated the experiment by preliminarily transfecting MCF10A cells with siRNA in order to inhibit E-cadherin release. Lipofectamine and siRNA were added to MCF10A cells for one hour before adding TGF-β1. The experiment was then performed as previously described (five days of TGF- β stimulation, then followed by myo-Ins addition for six hours). Transfection of siRNA significantly decreased E-cadherin release in controls as well as in cells treated with TGF-β1, myo-Ins, or both (Figure 10). Noticeably, the addition of myo-Ins to TGF-β1 treated cells was unable in restoring E-cadherin–β-catenin complexes, as well as cell-to-cell junctions. When we add myo-Ins, cells remain scattered and detached from each other. Moreover, cells were unable to recover their epithelial shape, maintaining a mesenchymal-like profile. In the meantime, β-catenin moved from the membrane to migrate into the cytosol, while E-cadherin density behind the cell membrane almost completely disappeared. Conclusively, silencing of E-cadherin almost completely antagonizes the “normalizing” effect exerted by myo-Ins upon cells committed toward EMT. Overall, such findings suggest that the reconstitution of E-cadherin–β-catenin structures is a mandatory step to enable myo-Ins to counteract the TGF-β1 effects.

## 4. Discussion

Epithelial cells show a well-structured and polarized architecture, enabling cells to interact with each other through specialized structures—mostly based on E-cadherin-forming bridges with β-catenin. This intertwined structure is severely compromised during EMT, which allows polarized cells to undergo multiple biochemical and structural changes that enable them to assume a mesenchymal cell phenotype, characterized by enhanced migratory capacity, invasiveness, elevated resistance to apoptosis, and greatly increased production of extracellular matrix (ECM) components.

However, EMT—being a system’s feature—cannot be assessed only by focusing on a few molecular markers. EMT represents a dramatic modification of cell behavior. Cells lose their main epithelial features while acquiring mesenchymal traits. In particular, the most impressive modifications usually observed involve the acquisition of migrating and invasive properties, characteristics that play a critical role in morphogenesis, as well as in cancer progression. Those changes require the cooperation of a large number of molecular and biophysical factors working in concert across different scales in space and time. Therefore, EMT would benefit by being described as a functional change in the biological properties of cells rather than focusing largely on changes in a few, readily monitored molecular markers. Definitely, as recommended by the Consensus Statement on EMT, “whenever it is experimentally feasible, EMT should be assessed through the combination of cellular properties and multiple molecular markers” [27].

Indeed, we observed that under the stimulation exerted by TGF-β1—a well-known EMT inducer—breast MCF10A cells dramatically transformed their morphology and lose the apical-basal polarity, while becoming progressively “detached” from each other. Consequently, “transformed” cells acquire high motility and invasiveness. This finding is an unequivocal confirmation of the EMT occurring in epithelial cells stressed by TGF-β1 addition.

To acquire a migrating–invasive phenotype, cells must primarily detach from each other. Within an organized tissue, cooperation among cells is chiefly assured by cell–cell interactions—including adherent junctions, tight junctions, and desmosomes. Additionally, this ensemble of connecting bridges is instrumental in facilitating signal transmission across the cell population. The maintaining of apical-basal polarity is needed to instruct a proper formation of cell junctions [28]. Consequently, loss of apical–basal polarity is often the first event that appears before we could observe the destabilization of adhesion complexes [29]. Moreover, several reports have highlighted that the presence of an actin-rich domain in adhering cells is a general phenomenon of polarizing cells [30]. In our experiments, we indeed noticed how the control cells displayed an apical domain enriched in F-actin, while in cells treated with TGF-β1, F-actin showed a uniform distribution and a flattened profile, suggesting that breast cells committed to EMT have lost their apical–basal polarity.

Cytoplasmic re-localization of E-cadherin and β-catenin via posttranscriptional regulation is a very early feature of EMT. The cortical tension ensured by the F-actin ring at the apical edge of the cell promotes E-cadherin anchoring to the actomyosin cytoskeleton, thereby increasing the clustering and stability of E-cadherin within contact zones. This finding outlines the importance of the actin cytoskeleton—arranged to assure a proper cell polarity—in stabilizing E-cadherin-based junctions. Indeed, if actin remodeling leads to increased stiffness, after exceeding a critical threshold level, the rate by which the E-cadherin-based contact expands in response to pulling forces from the cortex sharply drops, leading to a progressive reduction of cell-to-cell contact [31]. Indeed, cadherins at cell–cell contacts not only transduce forces between the contacting cells, they are also in turn “molded” by those forces to which they are subject, showing a nonmonotonic relationship between cell-cortex tension and cell–cell contact size [31]. Consequently, the inactivation of E-cadherin–β-catenin is a downstream event following polarity disruption [32].

As expected, we observed that, upon TGF-β1 stimulation, MCF10A cells show a significant reduction in E-cadherin distribution behind the cell membrane, while β-catenin almost disappears and translocates either in proximity or into the nucleus. Noticeably, E-cadherin progressively slows down while N-cadherin—a hallmark of EMT [33]—steadily increases. Following the loosening of cell-to-cell adhesions, cells acquire increased motility through a sequence of different steps involving several other structural factors, including cytoskeleton remodeling, integrins, and motor protein changes, just to mention a few. Specifically, a switch of intermediate filaments from keratin-18 to vimentin can facilitate cell migration [34]. Indeed, we observed a significant downregulation of keratin-18 and an impressive five-fold increase in vimentin release in transformed cells.

A discrete number of key EMT transcription factors are associated with EMT and often, when activated, they can initiate further modifications. Snai1, a strong inducer of EMT has a pivotal role in this context. Snai1 acts upon several factors during physiological morphogenesis and can induce EMT and invasiveness also in normal somatic cells, specifically through transcriptional repression of the gene-encoding E-cadherin [35]. Conversely, Snai1 silencing effectively suppresses tumor growth and invasiveness [36]. It is well established that TGF-β1 is a strong inducer of Snai1 expression [37]. As expected, we observed that following TGF-β1 addition. Snai1 steadily increases up to three-fold in cells undergoing EMT. Interestingly, this increase is associated with a parallel upsurge in PI3K levels. The PI3K pathway, although apparently not required for the activity of the basal Snai1 promoter, is nonetheless needed for Snai1 promoter activation during EMT. Moreover, TGF-β1 promotes motility through mechanisms involving the activation of PI3K and Snai1 [38]. Noticeably, PI3K activity is required to allow cell scattering and survival after TGF-β1 treatment [39]. Indeed, overexpression of PI3K leads to the activation of several pathways downstream—including pAkt, NF-kB, Ras, and Wnt/β-catenin. Overall, these remarks substantiate the fact that PI3K is currently considered a hallmark of EMT [40]. In our experiments, we indeed observed a dramatic increase in PI3K expression values following TGF-β1 treatment. Furthermore, EMT is associated with the breakdown of the extracellular matrix (ECM), a mandatory step that allows cells to invade into nearby normal tissues. A pivotal role in this setting is sustained by metalloproteinases. Matrix metalloproteinases actively participate in EMT by degrading and modifying the extracellular matrix as well as the cell-ECM and cell–cell contacts, facilitating the detachment of epithelial cells from the surrounding tissue [41]. As expected, TGF-β1 strongly induces the over-expression of MMPs expression [42]. In turn, MMPs, produced by either cancer cells or resident stroma cells, activate latent TGF-β1 in the extracellular matrix, thus reinforcing the overall transition. Finally, once transformed by EMT, clusters of cells initiate the production of increased levels of ECM components, such as collagen and fibronectin [43], as we observed in our experiments. The increased release of both collagen and fibronectin can further promote the transforming process by actively modifying the biophysical microenvironment in which cells are growing.

It is remarkable that in our model, under continued exposure to TGF-β1, cells began releasing increased concentrations of TGF-β1 itself. This finding is intriguing and, to our best knowledge, was never noticed before. However, a previous study showed that, in a three-dimensional context, urokinase receptor (uPAR)—a member of the extracellular proteolytic enzyme system sharing some metalloproteinase features—elicits an increased release of TGF-β1 from different cancerous cells [44]. Unfortunately, in our study, we did not plan to investigate uPAR levels. Therefore, to ascertain whether the increased release of MMPs following TGF-β1 exposure could promote TGF-β1 synthesis deserves a future investigation.

Amazingly, all these features were almost completely nullified when we added myo-Ins to TGF-β1 transformed cells. After a few hours, the cells restored a physiological E-cadherin–β-catenin profile, thus “reverting” their mesenchymal phenotype while cells lose both migrating and invasive properties. At the same time, PI3K, Snai1, N-cadherin, vimentin, and collagen were significantly downregulated, in some cases reaching the same levels expressed in control cells. The only exception was represented by fibronectin, which was unaffected by myo-Ins treatment. Overall, these results suggest that myo-Ins induces a significant mesenchymal-to-epithelial reversion. However, it is worthy of note that the “reverting” process is only partially achieved, given that some parameters—albeit dramatically reduced—did not recover their basal values. Likely, this may be ascribed to the fact that our investigation extended only six hours after myo-Ins treatment. It remains to be determined whether a more prolonged period of treatment would have resulted in a more complete reversion.

To test the hypothesis that E-cadherin re-expression occupies a central role in promoting MET under myo-Ins treatment, we investigated if the inhibition of E-cadherin synthesis through siRNA addition in cells conditioned with TGF-β1 could impair the inositol capability in reverting EMT. Indeed, cells treated with siRNA were unaffected by the subsequent addition of myo-Ins, as they remain detached from each other, preserving the mesenchymal phenotype acquired following TGF-β1 induction. The overall content of E-cadherin was persistently low, while β-catenin was distributed in the cytosol in spite of myo-Ins addition. This result highlights the pivotal role that the re-establishment of E-cadherin–β-catenin complexes plays in triggering MET. Results of the silencing experiment demonstrated that the reverting effect of myo-Ins relies on the possibility of restoring appropriate levels of E-cadherin levels and preserving the correct spatial distribution of E-cadherin–β-catenin complexes. Our data provide further confirmation that the re-expression of E-cadherin is a mandatory step to reverse EMT and the cancerous phenotype [45]. Strong evidence suggests that loss of E-cadherin-mediated cell adhesion is one rate-limiting step in the progression from adenoma to carcinoma [46]. Conversely, re-establishing the functional cadherin complex in tumor cell lines results in a reversion from an invasive to a benign epithelial phenotype [47]. Therefore, it is of relevant interest that a simple and safe molecule like inositol can efficiently restore such a critical parameter.

Myo-Ins is credited with inducing several beneficial effects as an anticancer agent [20]. Namely, myo-Ins has been shown to induce EMT reversion in a breast cancer cell model [48]. In that model, by treating cells with pharmacological doses of inositol, several EMT parameters were significantly up and downregulated.

Similar results have been provided herein, demonstrating that myo-Ins can induce a significant EMT reversion even in normal breast cells committed toward a mesenchymal phenotype upon TGF-β1 addition. As reported by several studies, EMT can be forced to “reverse” under specific molecular and environmental cues [49]. Moreover, the recent discovery that MET is required for transforming somatic cells into pluripotent stem cells [50] suggests that the intersection between EMT and MET is a fundamental crossroad for cell-fate decisions. The effectiveness of myo-Ins in enacting such a reversion is of relevant interest for understanding key factors involved in MET. Moreover, given its safe pharmacological profile, myo-Ins should be taken into consideration as a useful and promising therapeutic support for a wide array of pathological conditions, including inflammatory states and cancer. It is worth noting that myo-Ins belongs to the class of migrastatics-antimetastatic and anti-invasion Drugs, a new class of pharmaceutical agents that target the motility and invasiveness of cancer cells [51]. As advocated by a number of recent approaches, it is evident that drug-discovery efforts should be dichotomized into antiproliferative strategies, and those directed toward mechanisms related to motility, migration, and/or invasion (metastasis). Current treatment protocols are mostly centered on altered signaling pathways, which are specifically involved in driving abnormal cellular proliferation. However, as Solomon and colleagues have noted, “efforts should refocus on the deadliest characteristic of all malignant cancers, i.e., metastasis. It is unsettling that, to this end, there are no anti-metastatic drugs available to patients with solid tumors” [52]. Undoubtedly, critical assessment of these antimetastatic agents is urgently warranted, since they may define new options for the treatment of solid cancers.

## 5. Conclusions

The inflammatory condition can favor the emergence of cancer by inducing relevant modifications in cell architecture—ultimately leading to EMT—and in the surrounding microenvironment. The transition toward an epithelial-like phenotype is mandatory to facilitate the disentanglement of tissue structure and in promoting the acquisition of malignant features, including enhanced migrating and invasive properties. Therefore, counteracting the dissemination of precancerous cells is a critical target to prevent metastasis. Unfortunately, despite recent advances in this field, specific treatments able to target EMT are still lacking. Here, we provide strong evidence that a safe, natural polyol—the myo-Inositol—can efficiently counteract biochemical, genetic, and structural changes accompanying the induction of an inflammatory phenotype. Indeed, myo-Ins can normalize gene expression, cytoskeleton architecture, and many related pathways, thus inhibiting the invasiveness and migrating capability of a cell population challenged by TGF-β1 stimulation. Overall, myo-Ins can efficiently promote an almost complete mesenchymal-to-epithelial reversion. Notably, the re-establishment of E-cadherin–β-catenin based adhesion structures reveals this to be a critical step during phenotypic reversion, as they are instrumental in ensuring a tissue-like architecture. Moreover, myo-Ins was effective also in reducing collagen and metalloproteinases, thus “normalizing” the microenvironment. These results suggest that the acquisition of malignant traits can be significantly counteracted with natural compounds. The possibility of mitigating with natural molecules some cancerous consequences tied to inflammation deserves to be explored in depth, as this kind of approach can disclose unexpected perspectives in cancer prevention and treatment.

## Figures and Tables

**Figure 1 cancers-15-02317-f001:**
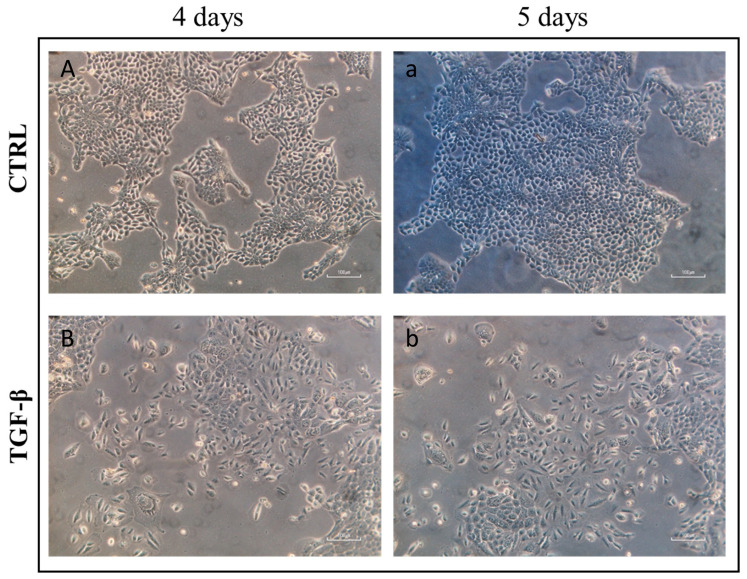
TGF-β1 triggers morphological changes in MCF-10A cells after five days of induction. Images on light microscopy of MCF-10A cells. In physiological conditions, epithelial cells are polarized, display a cuboidal-like morphology, and interact with each other through cell–cell junction (**A**,**a**). Upon TGF-β1 addition, cells lose cell–cell junction, and apico–basal polarity assumed a fibroblast-like morphology and became scattered over the entire area (**B**,**b**), thereby losing reciprocal connectivity and the tissue-like architecture.

**Figure 2 cancers-15-02317-f002:**
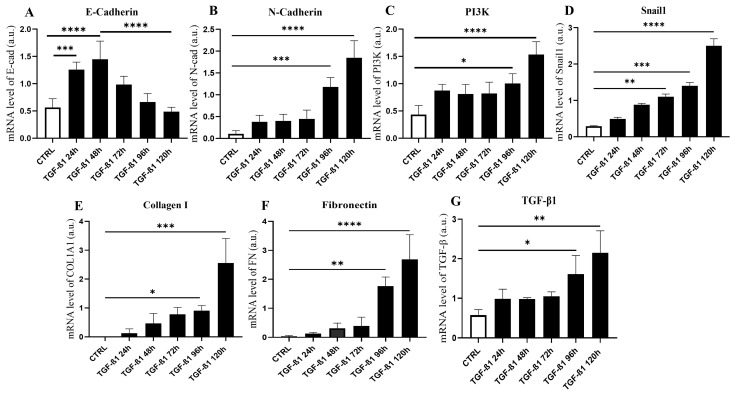
TGF-β1 triggers modulation of epithelial and mesenchymal marker expression. The mRNA levels of E-cadherin (**A**), N-cadherin (**B**), PI3K (**C**), Snai1 (**D**), Collagen (**E**), Fibronectin (**F**), and TGF-β (**G**) in the control and TGF-β1-treated MCF-10A cells at different time points. Following TGF-β1 treatment, noncancerous breast cells progressively lose the epithelial features, acquiring mesenchymal hallmarks, such as De Novo expression of N-cadherin, increased synthesis of Snai1, and reduced expression of E-cadherin. Noting that once the transformation process has reached a steady stae (i.e., 120 h), the system requires a very relevant increase in PI3K activity. Moreover, EMT is accompanied by a significant change in collagen and fibronectin release that can, in turn, modify the microenvironment architecture. Histograms are expressed as relative expressions of three independent experiments (mean ± SEM). GAPDH was used for housekeeping. One-way ANOVA statistical analysis: * *p* < 0.05; ** *p* < 0.01; *** *p* < 0.001; **** *p* < 0.0001.

**Figure 3 cancers-15-02317-f003:**
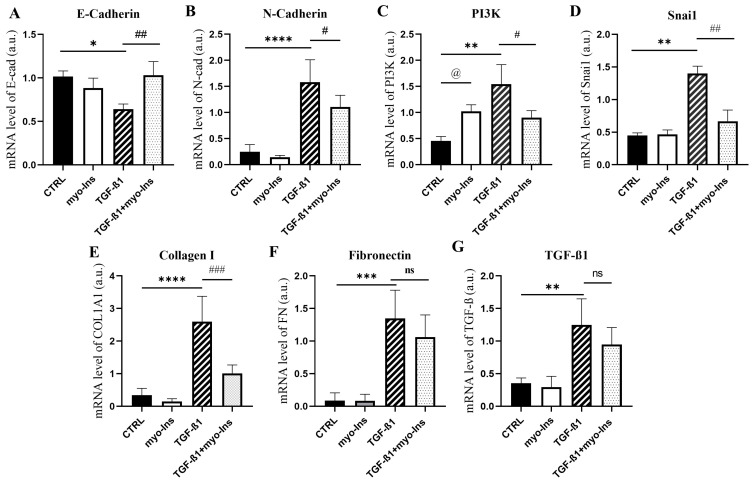
Myo-Ins downregulates the expression of mesenchymal parameters induced by TGF-β1. qPCR-evaluating levels of E-cadherin (**A**), N-cadherin (**B**), PI3K (**C**), Snail (**D**), Collagen (**E**), Fibronectin (**F**), and TGF-β1 (**G**) in MCF-10A cells treated with TGF-β1 alone or in combination with myo-Ins. The addition of myo-Ins induces, as early as after one hour, a significant reversion of all but one (fibronectin) of the investigated parameters. This result indicates that myo-Ins exerts a pleiotropic effect upon several pathways instrumental in fostering the mesenchymal–epithelial transition (MET). Histograms are expressed as relative expressions of three independent experiments (mean ± SEM). GAPDH was used for housekeeping. One-way ANOVA statistical analysis: * *p* < 0.05; ** *p* < 0.01; *** *p* < 0.001; **** *p* < 0.0001. # *p* < 0.05; ## *p* < 0.01; ### *p* < 0.001; @ *p* < 0.05; ns: no significance. Legend: *: TGF-β1 vs. CTRL; # TGF-β1+myo-Ins vs. TGF-β1; @: myo-Ins vs CTRL.

**Figure 4 cancers-15-02317-f004:**
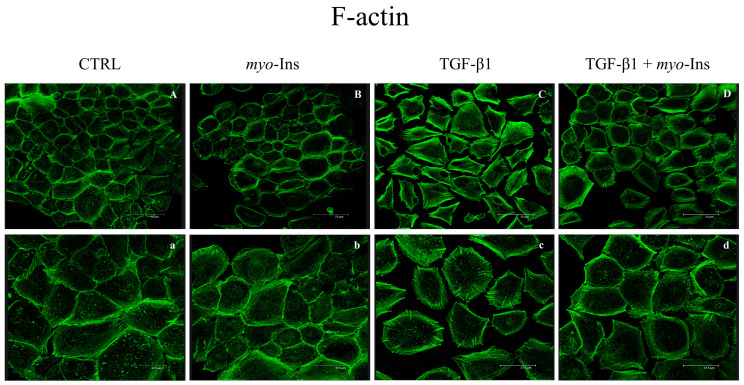
Myo-Ins modulates the F-actin remodeling induced by TGF-β1 in MCF-10A cells. Immunofluorescence analysis of F-actin in MCF-10A cells treated with TGF-β1 alone or with myo-Ins. In control cells (**A**,**a**) and in cells treated with myo-Ins alone (**B**,**b**), F-actin is organized in cortical position. Cells treated with TGF-β1 lose the typical basal-cell polarity, showing an isotropic distribution of actin behind the membrane (**C**,**c**). Presence of stress fibers increases within the cytosol. Lamellipodia and invadopodia are observed at the edge of transformed cells, which appeared scattered over the area, without clear connections to each other. The addition of myo-Ins almost completely reverts F-actin changes, re-establishing cell-to-cell contact (**D**,**d**). Images were scanned under a 40× objective. Scale bar of (**A**–**D**): 75 µm. (**a**–**d**) are a magnification of the same images.

**Figure 5 cancers-15-02317-f005:**
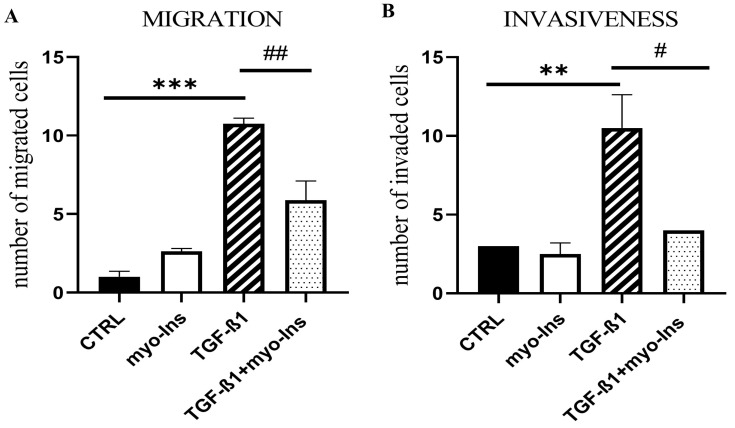
Myo-Ins inhibits motility and invasiveness. Quantification of the total number of migrating and invasive MCF-10A cells upon treatment with TGF-β alone or with myo-Ins following TGF-β stimulation. Transformed cells show an increase in motility (**A**) up to the height-fold compared to the control sample. Similarly, invasiveness is three-times higher in TGF-β1 treated cells than in the controls (**B**). Addition of myo-Ins significantly inhibits motility (<50% than in TGF-β1 treated cells) and almost completely nullifies the invasive capacity of transformed cells. Histograms indicate the mean value ± SD. ** *p* < 0.01; *** *p* < 0.001 (one-way ANOVA). # *p* < 0.05; ## *p* < 0.01. Legend: *: TGF-β1 vs. CTRL; # TGF-β1+myo-Ins vs. TGF-β1.

**Figure 6 cancers-15-02317-f006:**
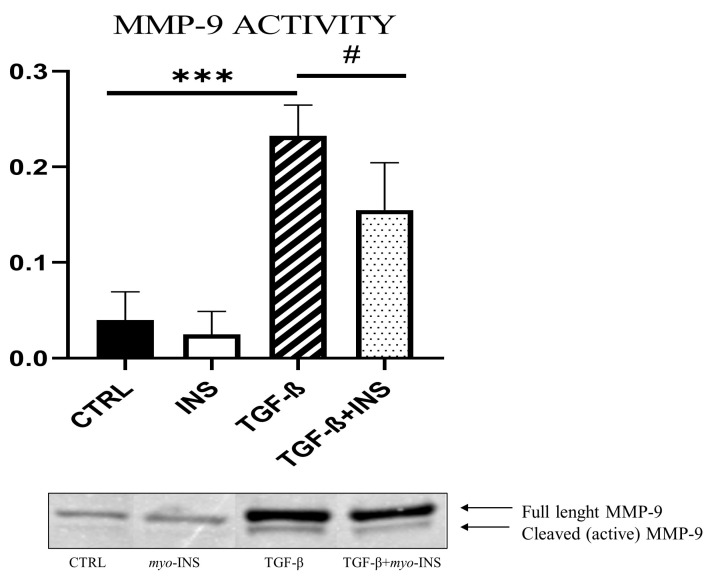
Myo-Ins decreases the proteolytic activity of MMP9. The histograms represent the MMP-9 activity in MCF-10A cells and in control MCF-10A cells treated with TGF-β1 alone or in combination with myo-Ins. Whereas TGF-β1 induce a five-fold increase in MMP-9 release, myo-Ins significantly mitigates this effect by ~40%. One-way ANOVA statistical analysis: *** *p* < 0.001. The uncropped blots are shown in Appendix A. # *p* < 0.05. Legend: *: TGF-β1 vs. CTRL; #: TGF-β1+myo-Ins vs. TGF-β1.

**Figure 7 cancers-15-02317-f007:**
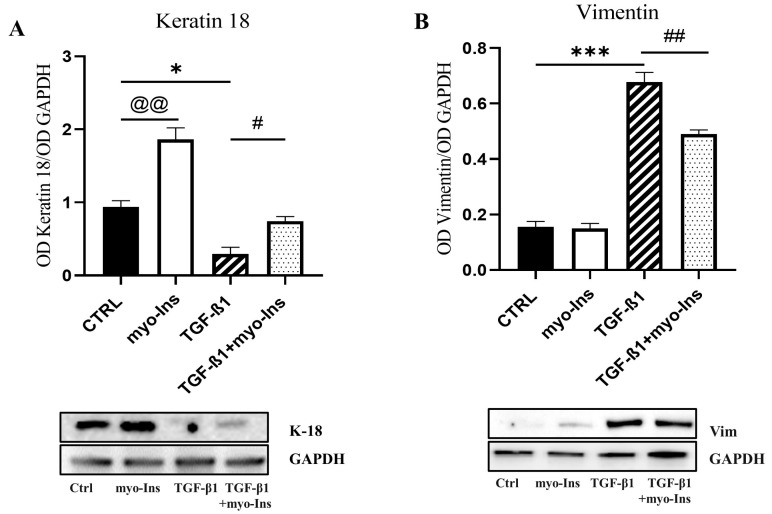
Myo-Ins modulates the expression of CSK microfilaments when added after five days of TGF-β1 induction. Graphs representing Western blot relative expression of keratin-18 (**A**) and vimentin (**B**) in MCF-10A cells treated with TGF-β1 alone or in combination with myo-Ins for six hours of three independent experiments (mean ± SEM). The addition of myo-Ins reverts almost completely the effects fostered by TGF-β1 upon intermediate filaments, by decreasing vimentin and augmenting keratin-18. Each sample data was normalized by referring to the relative expression of GAPDH. * *p* < 0.05; *** *p* < 0.001 (One-way ANOVA). The uncropped blots are shown in Appendix A. # *p* < 0.05; ## *p* < 0.01; @@ *p* < 0.01. Legend: *: TGF-β1 vs. CTRL; # TGF-β1+myo-Ins vs. TGF-β1; @: myo-Ins vs. CTRL.

**Figure 8 cancers-15-02317-f008:**
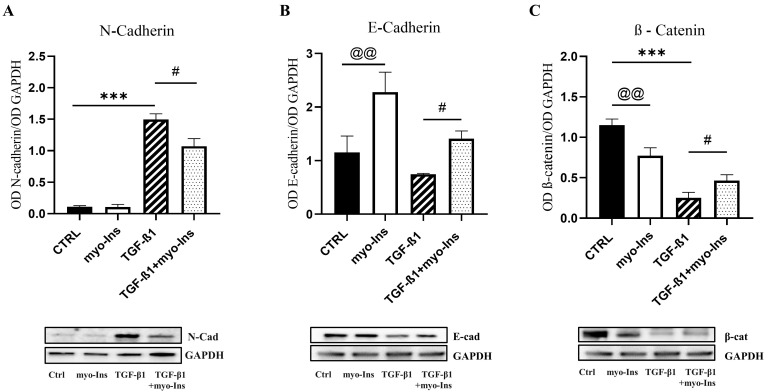
Myo-Ins regulates the expression of cell-to-cell junction markers when added after five days of TGF-β1 induction. Graphs representing Western blot relative expression of N-cadherin (**A**), E-cadherin (**B**), and β-catenin (**C**) in MCF-10A cells treated with TGF-β1 alone or in combination with myo-Ins for six hours of three independent experiments (mean ± SEM). The treatment with myo-Ins significantly (−40%) reduces the expression of N-cadherin, while promoting the expression of E-cadherin up to the level shown by the control samples. Moreover, myo-Ins rescues the TGF-β1-induced inhibition upon β-catenin. Although β-catenin increases up to two-fold when compared to transformed cells, the pre-TGF-β1 values are not reached after myo-Ins treatment. Each sample has been normalized by using the relative expression for GAPDH. *** *p* < 0.001 (One-way ANOVA). The uncropped blots are shown in Appendix A. # *p* < 0.05; @@ *p* < 0.01. Legend: *: TGF-β1 vs. CTRL; # TGF-β1+myo-Ins vs. TGF-β1; @: myo-Ins vs. CTRL.

**Figure 9 cancers-15-02317-f009:**
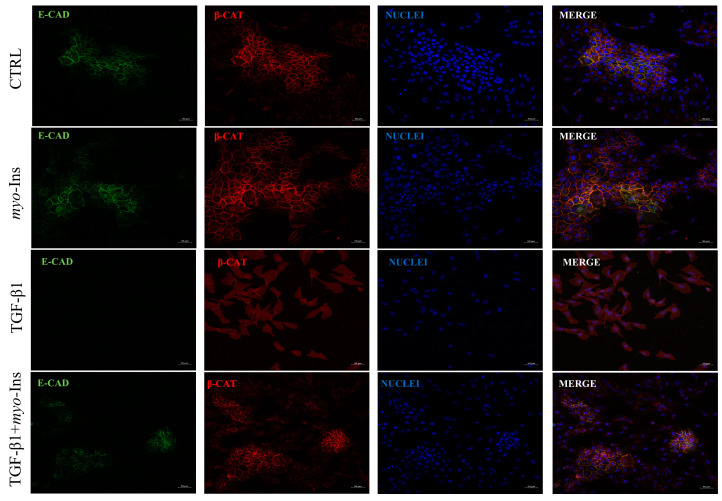
Analysis of E-cadherin and β-catenin distribution. Immunofluorescence analysis of E-cadherin (green) and β-catenin (red) in MCF-10A cells. The experimental conditions are depicted as control, myo-Ins treated, TGF-β1 treated, and cells treated with TGF-β1 plus myo-Ins. Nuclei were stained with TO-PRO3 (blue). The images were scanned under a 20× objective. In TGF-β1 treated samples, cells lose E-cadherin–β-catenin complexes, while β-catenin migrates in the cytosol and E-cadherin was dramatically downregulated. Cells are detached from each other and dispersed over the entire area. The addition of myo-Ins following TGF-β1 almost completely restores the physiological architecture of cell-to-cell contact with the reappearance of E-cadherin/β-catenin complexes. Cells recover their junctions and assume a compact, tissue-like profile, losing almost entirely stress fibers.

**Figure 10 cancers-15-02317-f010:**
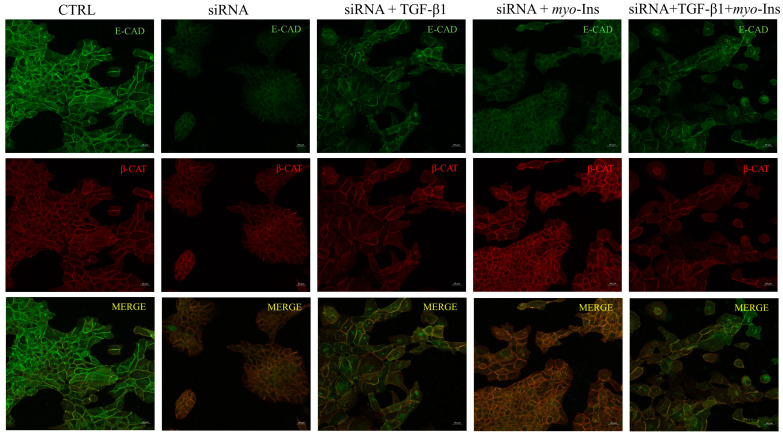
Analysis of E-cadherin and β-catenin distribution. Immunofluorescence analysis of E-cadherin (green) and β-catenin (red) in MCF-10A cells transfected with E-cadherin siRNA and then treated with myo-Ins, TGF-β1 alone, or a sequential association of the two factors. In contrast to the results obtained in the absence of transfection with E-cadherin siRNA, myo-Ins was unable to reverse the effects induced by TGF-β1. Restoration of E-cadherin–β-catenin complexes only marginally occurs, and cells remain detached and scattered all around. F-actin distribution is only partially improved, and stress fibers are still visible. Overall, images suggest that the reconstitution of E-cadherin–β-catenin structures is a mandatory step to achieve EMT reversion upon myo-Ins treatment. Images were scanned under a 20× objective.

## Data Availability

Data are contained within the article or the Appendix A. The data presented in this study are available.

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
