# Peer review of "Myo-Inositol Reverses TGF-β1-Induced EMT in MCF-10A Non-Tumorigenic Breast Cells"

_cancers, 2023, doi:10.3390/cancers15082317_

Round 1

Reviewer 1 Report

This is an extremely important study providing evidence for normalization of pre-cancerous or early tumor environment by a nont-toxic all natural nutritional supplement via EMT reversal with documented critical role of E-cadherin. This approach - non-toxic treatment of cancer via normalization of tissue microenvironment has potential to revolutionize cancer treatment. 

Minor revision:

Fig. 1 would benefit from contrast enhancement.

Reviewer 2 Report

cancers-2338986-peer-review:  Myo-inositol reverses TGF-b1-induced EMT in MCF10A non-tumorigenic breast cells.

This is a very strong paper addressing a comprehensive set of studies examining the capability of myo-inositol to reverse the Epithelial-Mesenchyme Transition (EMT) induced and sustained by TGF-b1 during inflammatory and cancer initiating processes.  The ntroductory literature review is appropriate for understanding the logic of the series of experiments conducted by the authors.  The complex Methodology and Results sections are clear (but see comments below), and the Discussion appropriately puts the data into cellular and physiological context based on appropriate referencing to the relevant published literature.

The choice of assays allowed for a comprehensive demonstration of the effects of TGF-b1, alone, and the almost total reversal of those effects when myo-inositol is added, to the cellular and subcellular integrity of the MCF10A cells.  The results potentially have both basic scientific as well as practical (pharmacological) implications.

My suggestions for editing are all meant to address clarity in the presentation of the data, mainly in the figures, as well as a couple of suggested edits in the Discussion..

Methods:

·      Line 195.  Please remove the symbols attached to various levels of significance.  Instead, these symbols and their interpretations should reside in figure legends.  The sentence should read, “… standard error of the mean, and p values of <.05 were considered statistically significant.”

Results:

·      All figures require appropriate legends.  While the results are described well in the text of the manuscript, legends make the reading of the figures much easier for the reader.  And, as suggested above, this is where the information about comparisons and statistical significance should be placed.

Discussion:

·      Line 453.   Reword to something less folksy, such as, ‘It remains to be determined whether a more prolonged period of treatment would have resulted in a more complete reversion.’

·      Lines 494-496.  This quote, from an unidentified source, has good content but is odd in format.  At least preface the quote with, ‘As Solomon and colleagues have noted, ….’

End notes:

·      Lines 499 on:  All of the information in these several sections needs to be provided.
